# Analysis of Heart-Rate Variability during Angioedema Attacks in Patients with Hereditary C1-Inhibitor Deficiency

**DOI:** 10.3390/ijerph18062900

**Published:** 2021-03-12

**Authors:** Francesca Perego, Beatrice De Maria, Maria Bova, Angelica Petraroli, Azzurra Marcelli Cesoni, Valeria De Grazia, Lorenza Chiara Zingale, Alberto Porta, Giuseppe Spadaro, Laura Adelaide Dalla Vecchia

**Affiliations:** 1Department of Medicine, IRCCS Istituti Clinici Scientifici Maugeri, 20138 Milan, Italy; azzurra.cesoni@icsmaugeri.it (A.M.C.); valeria.degrazia@icsmaugeri.it (V.D.G.); lorenza.zingale@icsmaugeri.it (L.C.Z.); laura.dallavecchia@icsmaugeri.it (L.A.D.V.); 2Bioengineering Laboratory, IRCCS Istituti Clinici Scientifici Maugeri, 20138 Milan, Italy; beatrice.demaria@icsmaugeri.it; 3Department of Translational Medical Sciences and Center for Basic and Clinical Immunology Research (CISI), University of Naples “Federico II”, 80138 Naples, Italy; bovamaria@virgilio.it (M.B.); petrarol@unina.it (A.P.); spadaro@unina.it (G.S.); 4Department of Biomedical Sciences for Health, University of Milan, 20133 Milan, Italy; alberto.porta@unimi.it; 5Department of Cardiothoracic, Vascular Anesthesia and Intensive Care, IRCCS Policlinico San Donato, San Donato Milanese, 20097 Milan, Italy

**Keywords:** C1-inhibitor hereditary angioedema, heart rate variability, cardiac neural control, autonomic nervous system, spectral analysis, multiday ECG monitoring, quality of life, burden of disease

## Abstract

C1-inhibitor hereditary angioedema (C1-INH-HAE) is a rare disease characterized by self-limiting edema associated with localized vasodilation due to increased levels of circulating bradykinin. C1-INH-HAE directly influences patients’ everyday lives, as attacks are unpredictable in frequency, severity, and the involved anatomical site. The autonomic nervous system could be involved in remission. The cardiac autonomic profile has not yet been evaluated during the attack or prodromal phases. In this study, a multiday continuous electrocardiogram was obtained in four C1-INH-HAE patients until attack occurrence. Power spectral heart rate variability (HRV) indices were computed over the 4 h preceding the attack and during the first 4 h of the attack in three patients. Increased vagal modulation of the sinus node was detected in the prodromal phase. This finding may reflect localized vasodilation mediated by the release of bradykinin. HRV analysis may furnish early markers of an impending angioedema attack, thereby helping to identify patients at higher risk of attack recurrence. In this perspective, it could assist in the timing, titration, and optimization of prophylactic therapy, and thus improve patients’ quality of life.

## 1. Introduction

Angioedema (AE) without wheals is a localized self-limiting edema associated with different mechanisms. The best-known form is hereditary angioedema (HAE) due to C1-inhibitor (C1-INH) deficiency, a rare disease with a prevalence of 1:65,000 in Italy [1]. Symptoms include swelling of the extremities, genitals, bowel mucosa, face, and upper airways, including the larynx. Laryngeal attacks, if untreated, can lead to death [2]. AE attacks are unpredictable and occur episodically upon release of the main mediator of attacks, namely, bradykinin, due to the hyperactivation of the contact system lacking its main control protein, C1-INH [2]. The overall result is an impairment of endothelial function associated with increased vascular permeability [3]. The release of bradykinin occurs locally and unpredictably, at times facilitated by trauma and other triggers such as stress [4,5].

The burden of disease for patients with HAE is substantial and well-documented [6]. C1-inhibitor hereditary angioedema (C1-INH-HAE) not only causes substantial short-term disability associated with attacks, but may also lead to persistent anxiety between episodes.

Despite full physical recovery between attacks, patients often experience continual emotional impairment and reduced quality of life [7].

HAE has negative impact on education and career. Loss of productivity due to absenteeism from work or attendance at educational activities for patients and caregivers has significant negative economic impact and increases stress during and between attacks. Missed opportunities in education and career development are common [8].

The financial burden attributable to HAE can be divided into direct and indirect costs. Indirect costs are estimated to be $16,108 per year, including those related to reduced workplace productivity, reduced income, missed work and travel, and childcare [9].

A survey conducted in the United States reported that 57% of HAE patients felt that their career advancement was hindered, 69% felt that they could not consider certain types of jobs due to their disease, 63% felt HAE impacted their career choices, 40% did not progress as far in school as desired, 48% felt that their educational advancement was hindered, and 55% felt that their educational choices were limited [7].

The regulation of vascular permeability is responsible for the clinical evolution of HAE attacks [4]. Given the link between the effects of bradykinin on the endothelium and the modulation of the autonomic nervous system (ANS) on vessels, the ANS can modulate the severity of attacks.

In C1-INH-HAE patients, the ANS has a role in the regulation of vascular permeability, for example, via the baroreflex mechanism [10,11]. Sympathetic-nervous-system inhibition by α2 agonist clonidine reduces microvascular permeability in endotoxemic animals, suggesting that antagonizing the sympathetic nervous system may prove beneficial in stabilizing capillary leakage during inflammation. Similarly, the vagus nerve has a protective role in models of inflammation such as ischemia–reperfusion injury [12]. The parasympathetic tone, acting on B_2_ receptors in nucleus ambiguous, can also be modulated by bradykinin [13]. Moreover, in idiopathic systemic capillary leak syndrome, a disease characterized by unexplained episodes of plasma extravasation dealing severe hypovolemic shock, heart-rate variability (HRV) analysis showed that, during the hyperacute phase of shock, both sympathetic and vagal modulations were totally absent, mimicking the pattern of a denervated transplanted heart [14]. The ANS was investigated by means of power spectral analysis of HRV in C1-INH-HAE patients during a remission period [15]. The main finding was that of increased sympathetic activation at rest and a blunted ANS response during the orthostatic challenge. However, the role of the ANS during AE attacks or its prodromal phase was not explored.

The ANS might play an either direct or indirect role in the generation of attacks. Our hypothesis regarding the behavior of ANS during attacks was not preoriented to a specific pattern. The aim of this work was to evaluate the cardiac ANS by means of HRV, i.e., analysis of the spontaneous fluctuations of heart period usually approximated by the RR interval [16] in C1-INH-HAE patients during an attack and in the hours preceding. Preliminary results were presented at the 11th Conference of the European Study Group on Cardiovascular Oscillations [17].

## 2. Methods

### 2.1. Population Characteristics

Four C1-INH-HAE patients were enrolled in the study. Below, we provide the clinical and demographic characteristics of the patients.

Patient 1: A 38 year old female with C1-INH-HAE Type 1. HAE was diagnosed at the age of 26. She suffers from recurrent (once a week) episodes of abdominal and peripheral AE, and sporadic attacks of glottal edema. She took long-term prophylactic therapy (danazol 50 mg every other day) for 2 years prior to the study, and intravenous (IV) C1-INH (at a dosage of 1000 IU) for the treatment of acute attacks. She also suffers from allergic rhinitis and is treated with specific immunotherapy for grasses with good symptom control. On Day 6 of recording, the patient had a mild cutaneous AE attack at approximately 9 a.m., which was relieved by self-administered IV C1-INH within 2 h.

Patient 2: A 31 year old male with C1-INH-HAE Type 1. HAE was diagnosed at the age of 12. He suffers from recurrent (one every 4 to 5 days) episodes of peripheral AE and abdominal attacks, and several laryngeal attacks treated with IV C1-INH. Until a year ago, the patient was on subcutaneous therapy with icatibant for treatment of the acute attack with a relapse of symptoms occurring after 12–24 h. The patient used IV C1-INH (at a dosage of 2000 IU) to terminate attacks. No long-term prophylactic therapy was in use. He is also a healthy carrier of the hepatitis C virus. On Day 1 of recording, the patient had a moderate cutaneous AE attack accompanied by stomach pain at approximately 12 a.m., and was treated with IV C1-INH to beneficial effect within 10 h. On Day 4 of recording at 10 a.m., the patient had a moderate cutaneous attack lasting for approximately 12 h. The attack was treated with self-administered IV C1-INH to beneficial effect within 10 h. The Day 4 attack was considered for analysis based on ECG quality.

Patient 3: A 46 year old female with C1-INH-HAE Type 1. HAE was diagnosed at the age of 8. She is otherwise healthy. During Day 2 of recording, the patient had a moderate abdominal HAE attack at approximately 2 p.m. lasting for 12 h.

Patient 4: A 53 year old female and the mother of Patient 2, with C1-INH-HAE type 1. HAE was diagnosed at the age of 33. She suffers from recurrent episodes of abdominal and peripheral angioedema. Over the last 13 years, the number of attacks increased to a frequency of one attack per week, with several attacks of glottal edema necessitating IV C1-INH at a dosage of 1500 IU. The patient is free from other diseases. She was excluded from further analysis due to the presence of several ectopic beats constituting more than 5% of the total beats.

### 2.2. Experiment Protocol

All patients underwent a multiday ambulatory ECG recording until attack occurrence using a 3 lead multiday ECG recorder (eMotion Faros 360°, MegaElectronics Ltd., Finland; Sylco srl, Monza, Italy). ECGs were sampled at 125 Hz.

ECG recorders were positioned at enrollment, and patients were invited to wear them until the occurrence and complete resolution of the first attack. Thereby, the duration of the recording differed from patient to patient and depended on the timing of attacks. We provide details of the analyzed periods of recordings below. No limitations on daily activities were imposed. Patients were asked to maintain a diary with all relevant daily and nocturnal activities throughout the whole period of recording, including the characteristics of any attacks, the day and time of their onset, and their severity, localization, date and time of treatment, and details of the regression of symptoms.

The study adhered to the principles of the Declaration of Helsinki, and each subject signed a written informed consent prior to ECG recording. The study was approved by the IRCCS Maugeri Ethical Committee (approval number: CE2303; date of approval: 14 May 2019).

### 2.3. Time-Series Extraction and Selection of Segments for Analysis

ECG from lead II was selected for analyses due to the better signal-to-noise ratio. The heart period was approximated as the temporal distance between two consecutive R-wave peaks (RR) and is expressed in ms. R-wave peaks were detected by an automatic algorithm exploiting a method on the basis of a threshold on the first derivative and fixing the R-wave peak by means of parabolic interpolation [18]. Detection of the R-wave peaks was manually checked to avoid misidentifications. In the presence of ectopic beats or artifacts, correction by means of cubic spline interpolation was implemented. No more than 5% of the total beats were corrected.

Segments for power spectral analysis were selected on the basis of the occurrence of the attack for each patient. We considered the 4 h preceding the attack (PRE) and the first 4 h following the onset of the attack (ONGOING) on the day of the attack (Day_Attack), and the same circadian periods on the day before (Day_BEFORE_Attack). Therefore, analyses of Day_Attack and Day_BEFORE_Attack were comparable in terms of circadian phase within the same subject. The RR series during the selected periods were extracted and analyzed. Time and frequency domain analysis was iterated over frames of 250 consecutive RR values with an overlap of 200 values [19]. The median of the distribution of the computed markers was taken as representative, and exploited in subsequent statistical analysis [19]. The RR mean (μ_RR_) was computed over the original series and expressed in ms, while RR variance (σ^2^_RR_) and spectral markers were calculated after linear detrending. σ^2^_RR_ is expressed in ms^2^.

### 2.4. Power Spectral Analysis

Parametric power spectral analysis was performed on the selected segments. The RR series was described by an autoregressive model of which the order was chosen according to Akaike’s information criterion. The power spectral density was decomposed into power spectral components. The sum of the power spectral components of which the central frequency dropped in the high-frequency (HF, 0.15–0.4 Hz) band represented the absolute power of RR series in the HF band (HFa_RR_) and is expressed in ms^2^ [16]. HFa_RR_ is also expressed in normalized units (HFnu_RR_), namely, the absolute value divided by the σ^2^_RR_ minus the RR power computed in the very low frequency band (0.0–0.4 Hz) and multiplied by 100 [16]. HFa_RR_ and HFnu_RR_ were taken as indices of the cardiac vagal modulation directed to the sinus node [20,21].

### 2.5. Statistical Analysis

Two-way repeated measures analysis of variance (Holm–Sidak test for multiple comparisons) was performed to assess the effect of the attack within the same patient by considering the circadian phase. The possible confounding influence of the circadian phase was minimized by the comparison of data on the day of the attack with those collected in the same period on a different day in which an attack was not observed. Data were analyzed by separately considering the hour-by-hour indices in the PRE and ONGOING phases, and after pooling the data together regardless of time. Analysis was individually carried out for each patient and after pooling together all patient data. Data are presented as mean ± standard deviation. Statistical analysis was carried out using a commercial statistical program (Sigmaplot, ver.11.0, Systat Software, San Jose, CA, USA). A *p* value < 0.05 was considered significant.

## 3. Results

Figure 1 shows the results of μ_RR_, σ^2^_RR_, HFa_RR_, and HFnu_RR_ in Day_Attack (white bars) and Day_BEFORE_Attack (black bars) over the PRE (i.e., the 4 h preceding the attack) and ONGOING (i.e., the first 4 h of the attack) phases in the three analyzed patients. HFnu_RR_ was significantly higher in Day_Attack compared to Day_BEFORE_Attack in the PRE phase. No other significant variations between Day_Attack and Day_BEFORE_Attack or between PRE and ONGOING phases were observed for any of the considered indices.

Figure 2 shows the results of μ_RR_, σ^2^_RR_, HFa_RR_, and HFnu_RR_ in Day_Attack (white bars) and Day_BEFORE_Attack (black bars) over the PRE and ONGOING phases for each patient. In Patient 1, all indices were lower in the ONGOING phase compared to the PRE phase in both Day_Attack and Day_BEFORE_Attack, except for Hfnu_RR_ during Day_BEFORE_Attack. In Patient 2, μ_RR_, HFa_RR_, and HFnu_RR_ were higher in the PRE phase in Day_Attack compared to Day_BEFORE_Attack and, as a consequence, those indices were lower in the ONGOING phase compared to the PRE phase in Day_Attack. In Patient 3, all indices were lower in the ONGOING phase with respect to the PRE phase, indpendently of the considered day except for σ^2^_RR_. In this patient, HFnu_RR_ was higher during the PRE phase in Day_Attack compared to Day_BEFORE_Attack.

Figure 3 and Figure 4 show μ_RR_, σ^2^_RR_, HFa_RR_, and HFnu_RR,_ which were computed hour-by-hour in the PRE (i.e., 4 h preceding the HAE attack) and ONGOING (i.e., the first 4 h of the attack) phases of Day_Attack (white bars), and are presented alongside the corresponding time periods for Day_BEFORE_Attack. Figure 3 refers to the cohort of patients, and Figure 4 represents the trend of each patient.

## 4. Discussion

To our knowledge, this is the first study that evaluates heart-rate variability parameters during an AE attack and its prodromal phase in patients with C1-INH-HAE.

In this case series, patients were characterized by vagal hyperactivity during the prodromal phase of an acute attack. In fact, index HFnu_RR_ was significantly higher during the 4 h preceding the AE attack compared to the ongoing attack itself, and compared to the corresponding hours of the day without an attack.

This increased vagal modulation directed to the sinus node may reflect the localized vasodilation that is mediated by the release of the known main mediator of the attack, namely, endogenous nonapeptide bradykinin [4]. The role of endogenous mediators and expressed receptors on the cell surface in the development of angioedema attacks is largely studied and extensively described [2].

However, the mechanisms of possible triggers leading to an outbreak of an angioedema attack are still unknown and unpredictable. In addition, whether the increased vagal modulation detected in the patients of this study in the hours preceding the attack represents a mere epiphenomenon rather than a causative factor in the development of an AE attack remains to be established.

Bradykinin acts on cells and cardiac myocytes with vasodilator and cardioprotective effects [22]. Its effects on cardiovascular regulation are little-known. Bradykinin modulates the parasympathetic tone acting on B_2_ receptors in nucleus ambiguous [13,23]. However, the physiologic and pathophysiologic significance of such interaction is yet to be defined, although a role for bradykinin is suggested in some conditions such as hypertension and diabetes [24,25]. Similarly, the involvement of the ANS in C1-INH-HAE is still largely unexplained. The role of cardiovascular neural regulation in the pathophysiological mechanisms of C1-INH-HAE was recently demonstrated [15]. The increased sympathetic modulation seen in basal conditions in C1-INH-HAE patients with their impaired capability to respond to a sympathoexcitatory stimulus could represent the result of a complex central integrative process, as proposed in numerous conditions [26]. In keeping with our results, Wu et al. found increased circulating levels of cleaved kininogen, a breakdown product created when plasma kallikrein releases bradykinin [15]. Both findings are, in turn, corroborated by increasing evidence of the link between the chronic inflammation and regulation of the nervous system [27]. From this perspective, further studies are needed to elucidate how neural regulation is associated with the pathogenesis of C1-INH-HAE.

Clinical manifestations of HAE exhibit high intra- and interindividual variability, and angioedema attacks occur with unpredictable frequency and severity. The high variability and unpredictability of C1-INH-HAE manifestations affect patients’ daily lives and deeply influence their education and careers. As first observed in this small case series, HRV parameters may predict the occurrence of attacks. This has important clinical implications for identifying patients at risk of an impending attack, and for therapeutic guidance.

The identification of early markers, such as HRV parameters, could play a crucial role in helping patients to recognize an attack before the appearance of symptoms, and to manage it in an effective and timely manner.

## 5. Limitations and Perspectives

The main limitation of the present study is the small number of included patients. This aspect precludes carrying out adequate statistical analysis with appropriate power and sample size. However, due to the peculiar experimental conditions (i.e., the recording of patients during an acute HAE attack) and the total lack of data about ANS modulation during an attack in this rare disease, this pilot study provides some insights that may guide researchers in designing future studies. Indeed, this explorative study suggests that C1-INH-HAE patients did not mimic the pattern of a denervated transplanted heart. However, future studies should investigate in detail the extent, type, and direction of the ANS contribution to an HAE attack, including measurements of arterial blood pressure variability, in order to investigate the concurrent roles of both cardiac and vascular modulations.

Further research would also deepen the understanding of potential gender differences and gender-age-related characteristics since the development of attacks is linked to estrogenic status [28].

## 6. Conclusions

HRV analysis extended to multiday ECG recordings may furnish the early markers of an angioedema attack. Characterization of autonomic cardiac regulation in these patients may also help to identify an impending attack. From this perspective, HRV analysis could assist in the timing, titration, and optimization of prophylactic therapy. This new approach could enable patients to live a normal life, and lead to improved education and career opportunities.

## Figures and Tables

**Figure 1 ijerph-18-02900-f001:**
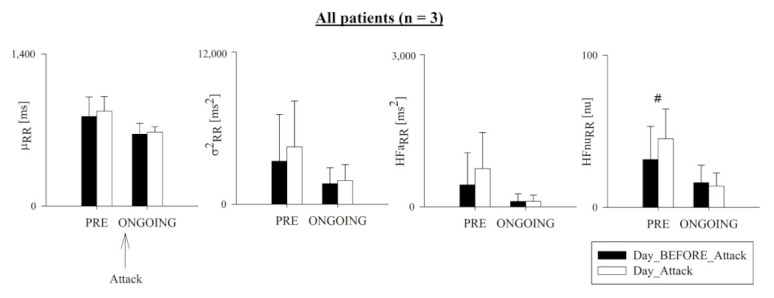
Pooled time- and frequency-domain heart-rate variability (HRV) markers in all patients (n = 3). Bar graphs show RR mean (μ_RR_), RR variance (σ^2^_RR_), and the absolute (HFa_RR_) and normalized (HFnu_RR_) power in the high-frequency (HF) band. Data in the 4 h preceding (PRE) and the first 4 h following the onset of the hereditary angioedema (HAE) attack (ONGOING) were pooled together on the day of the attack (Day_Attack, white bars) and compared with data during the same phase of the circadian rhythm on the day before (Day_BEFORE_Attack, black bars). Data are relevant to all patients and presented as mean ± standard deviation. Symbol # indicates *p* < 0.05 Day_Attack vs. Day_BEFORE_Attack.

**Figure 2 ijerph-18-02900-f002:**
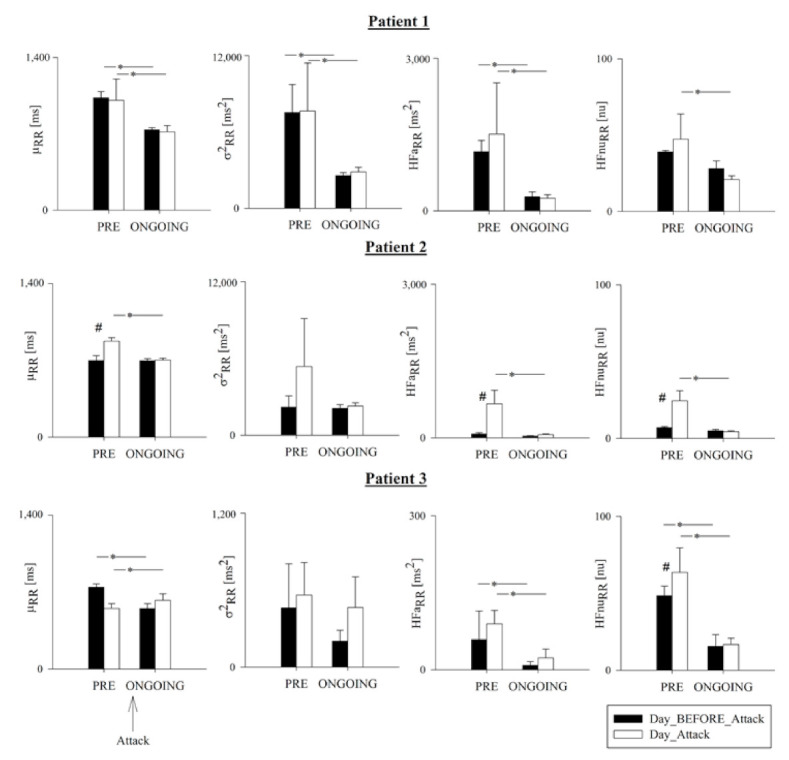
Pooled time- and frequency-domain HRV markers in Patients 1, 2 and 3. Bar graphs show RR mean (μ_RR_), RR variance (σ^2^_RR_), and absolute (HFa_RR_) and normalized (HFnu_RR_) power in the HF band. Hour-by-hour data in the 4 h preceding (PRE) and in the first 4 h following the onset of the HAE attack (ONGOING) were pooled together on the day of the attack (Day_Attack, white bars) and compared with data during the same phase of the circadian rhythm on the day before (Day_BEFORE_Attack, black bars). Data pertain to Patient 1 (upper panels), Patient 2 (middle panels), and Patient 3 (lower panels), and are presented as mean ± standard deviation. Symbol # indicates *p* < 0.05 Day_Attack vs. Day_BEFORE_Attack and the symbol * indicates *p* < 0.05 PRE vs. ONGOING.

**Figure 3 ijerph-18-02900-f003:**
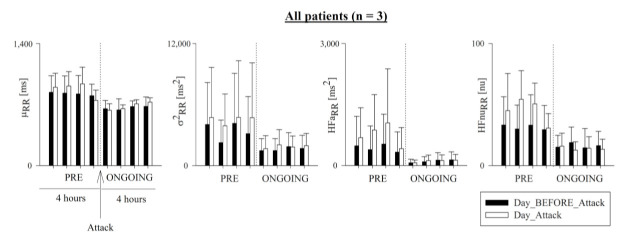
Temporal evolution of time- and frequency-domain HRV markers in all patients. Bar graphs show RR mean (μ_RR_), RR variance (σ^2^_RR_), and absolute (HFa_RR_) and normalized (HFnu_RR_) power in the HF band in all patients. Indices were calculated hour-by-hour in the 4 h preceding (PRE) and the first 4 h following the onset of the HAE attack (ONGOING) on the day of the attack (Day_Attack, white bars), and compared to data during the same circadian phase on the day before (Day_BEFORE_Attack, black bars). Results shown as mean ± standard deviation.

**Figure 4 ijerph-18-02900-f004:**
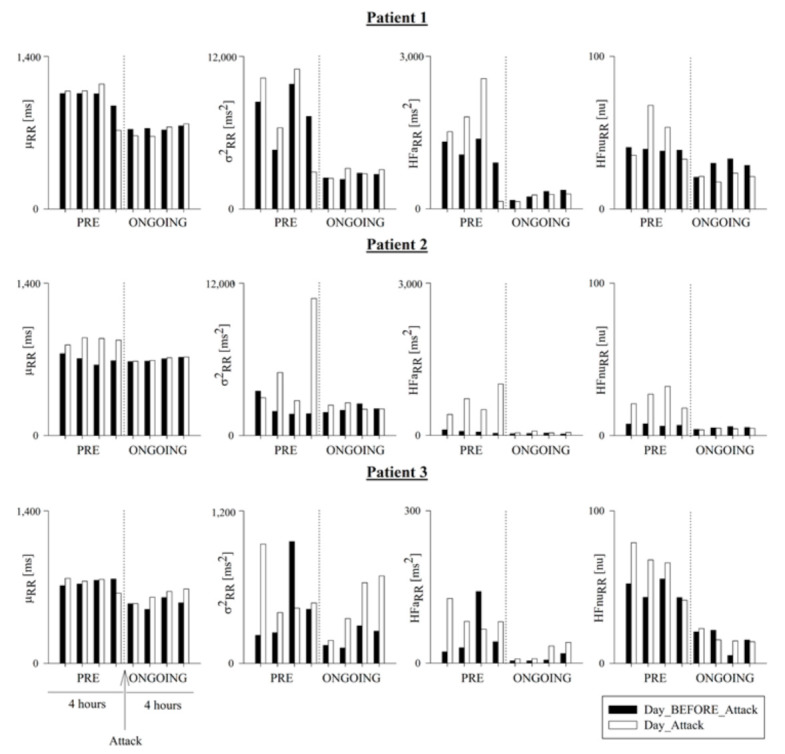
Temporal evolution of time and frequency domain HRV markers in Patients 1, 2 and 3. Bar graphs show the RR mean (μ_RR_), RR variance (σ^2^_RR_), and the absolute (HFa_RR_) and normalized (HFnu_RR_) power in the HF band in Patient 1 (upper panels), Patient 2 (middle panels), and Patient 3 (lower panels). Indices were calculated hour-by-hour in the 4 h preceding (PRE) and the first 4 h following the onset of the HAE attack (ONGOING) on the day of the attack (Day_Attack, white bars) and compared to data during the same circadian phase on the day before (Day_BEFORE_Attack, black bars). Results are shown as mean ± standard deviation.

## Data Availability

Data will be made available upon reasonable request to the corresponding author.

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
