# Peer review of "Analysis of Heart-Rate Variability during Angioedema Attacks in Patients with Hereditary C1-Inhibitor Deficiency"

_ijerph, 2021, doi:10.3390/ijerph18062900_

Round 1

Reviewer 1 Report

The following study aimed to evaluate the cardiac ANS by means of HRV in C1-INH-HAE patients during an attack and in the hours preceding. the study is well written has novelty, but major concerns must to be addressed.

Introduction:
The introduction is well written, and the putative underlying mechanisms of autonomic nervous system and C1-INH-HAE disease are quite interesting. However, I missed the hypothesis that the current study was undertaken to tested. For the authors, the hyphothesis is that C1-INH-HAE patients will be showed a classic response of many other disease (i.e., shyphatetic predominance and vagal withdrawal) or mimics the pattern of denervated transplanted heart patients? 

Methods

The experimental protocol should be better described, providing information such as body position, duration and period of the day of recording, preparations at the day and/or hours before the test. Please, describe all procedures that should be clear for readers.

As a major concern, how the authors reach the statistical power to perform Two-way repeated measures analysis of variance in the sample of four patients? Please show the calculated sample size for the outcome variables as the effect size for all variables.

Results and Discussion

Is there any HRV data from a control group?

Is there any data from autonomic tests (i.e., orthostatism, valsalva maneuver, cold pressor test ...)?

Please avoid the term "activity" when authors were referring to HRV indexes. The term "modulation" is better to describe the spontaneous fluctuations  on heart dynamics from ANS.

The discussion is well stated and the mechanisms regarding bradykinin and the overmodulation from vagal-mediated HRV is very interesting. However, is the blunted vagal-mediated HRV in C1-INH-HAE the primary or alternative hypothesis of the current study?

Did the study have no limitations? Please describe the potential limitations of the current study.

Author Response

Reviewer 1

Comments and Suggestions for Authors

The following study aimed to evaluate the cardiac ANS by means of HRV in C1-INH-HAE patients during an attack and in the hours preceding. the study is well written has novelty, but major concerns must to be addressed.

  1. Introduction: The introduction is well written, and the putative underlying mechanisms of autonomic nervous system and C1-INH-HAE disease are quite interesting. However, I missed the hypothesis that the current study was undertaken to tested. For the authors, the hyphothesis is that C1-INH-HAE patients will be showed a classic response of many other disease (i.e., shyphatetic predominance and vagal withdrawal) or mimics the pattern of denervated transplanted heart patients? 
  2. Methods: The experimental protocol should be better described, providing information such as body position, duration and period of the day of recording, preparations at the day and/or hours before the test. Please, describe all procedures that should be clear for readers.
  3. As a major concern, how the authors reach the statistical power to perform Two-way repeated measures analysis of variance in the sample of four patients? Please show the calculated sample size for the outcome variables as the effect size for all variables.
  4. Results and Discussion. Is there any HRV data from a control group? Is there any data from autonomic tests (i.e., orthostatism, valsalva maneuver, cold pressor test ...)?
  5. Please avoid the term "activity" when authors were referring to HRV indexes. The term "modulation" is better to describe the spontaneous fluctuations  on heart dynamics from ANS.
  6. The discussion is well stated and the mechanisms regarding bradykinin and the overmodulation from vagal-mediated HRV is very interesting. However, is the blunted vagal-mediated HRV in C1-INH-HAE the primary or alternative hypothesis of the current study?  
  7. Did the study have no limitations? Please describe the potential limitations of the current study.

Response to reviewer 1

We would like to thank the Reviewer 1 for having carefully read our manuscript and for his/her helpful suggestions.

  1. Thank you very much for the positive remarks. Regarding our hypothesis, taken in mind that in these patients, during a remission period, an increased sympathetic activation at rest and a blunted ANS response during the orthostatic challenge was found before, we hypothesized an ANS involvement before and during the AE attack or its prodromal phase. We now better explain this in the “Introduction” from line 64 to 67: “The regulation of vascular permeability is responsible for the clinical evolution of the attack [4]. Given the link between the effects of bradykinin on the endothelium and the modulation of the autonomic nervous system (ANS) on the vessels, ANS could modulate the severity of the attack.” and in lines 84-86: “We hypothesized that the ANS might play a role, either direct or indirect, in the generation of the attack. Our hypothesis on the behaviour of the ANS during the attack was not pre-oriented to a specific pattern.”
  2. Thank you. We improved the protocol description (see lines 131-138 and line 143)
  3. We agree with the reviewer that the statistical analysis is a limitation of the study (we added a comment in “Limitations and perspectives”, see lines 283-287). Unfortunately, the lack of data on this topic makes impossible a rigorous computation of the sample size at the level of the population. Conversely, we followed an individual approach over indices pooled together in PRE and POST phases in a specific subject. We remark that this pooling collected all the analyses carried out sequentially within a four-hour period in PRE and POST phases in each subject (Fig.2). However, we are aware that the small size of the group prevents any general statement at the level of the population, as suggested by Fig.1. Thus, we consider our findings as interesting preliminary results helpful to set further and more extensive applications.
  4. The purpose of the study was to evaluate the cardiac ANS in each C1-INH-HAE patient during an attack and in the hours preceding compared to a day without an attack. Thus, the focus was the attack itself during routine daily activities. Due to the specific design of the study protocol, we could not perform any additional test. A comparison between C1-INH-HAE patients and healthy subjects is certainly an excellent point to address in future studies. Testing these patients with different manoeuvres in controlled conditions would add important information about their cardiovascular autonomic profile. We are grateful for the suggestion.
  5. Thank you, correct. We changed the text, as suggested (line 252).
  6. This is a very important question. Unfortunately, the methods used in this study are not adequate to provide a unique interpretation of the data. Indeed, the lack of data about the vascular neural modulation does not allow exploring the whole mechanism. This point has been stressed in the “Limitations and perspectives” (lines 289-292). However, we believe that the data of the present study represent the basis for future protocols addressing more specifically the pathophysiology of the attack, in which the arterial pressure signal and/or other vascular oscillations will be included.
  7. Yes, we agree that there are limitations, we added a paragraph with the limitations of the study (lines 282-295).

Reviewer 2 Report

Abstract-

Although 4 patients were studied, as indicated in the methods, patient 4 was excluded.  Therefore, only data from 3 patients is reported. Please clarify in the abstract.

Please provide reference for the following statement: “AE attacks are unpredictable and occur episodically upon release of the main mediator of the attack, namely bradykinin, resulting from hyperactivation of the contact system lacking its main control protein .”

For the statement “Indirect costs have been estimated at $16,108, including reduced work- 56 place productivity, reduced income, missed work and travel, and lack of childcare [9],” please indicate if this is yearly costs or per lifetime, etc.

Suggest adding a few sentences or paragraph in the introduction to highlight that bradykinin modulates cardiac vagal tone.

A few references of interest:

https://www.ncbi.nlm.nih.gov/pmc/articles/PMC5798458/

https://pubmed.ncbi.nlm.nih.gov/9277549/

For this paragraph “The aim of this work was to evaluate the cardiac ANS by means of HRV, i.e. the 78 analysis of the spontaneous fluctuations of heart period, usually approximated by the RR 79 interval [15], in C1-INH-HAE patients during an attack and in the hours preceding”, please provide a sentence why that might be important to understand in patients with C1-INH-HAE. 

Results–

Figure 1, please indicate/clarify that the data are pooled from the average of multiple measurements from 3 individuals.  In other words, please indicate the n value or provide the Ftest numbers.

In Figure 2, it is interesting that the male patient showed a different pattern compared to the females.  Although it is limited in number, it is something the authors may wish to discuss.

Figure 2, please also list the number of measurements that were averaged per patient to generate the graphs.

Discussion—

As highlighted above, include a few sentences on potential sex differences that may be present and implications would strengthen the discussion.

Providing a paragraph highlighting the limitations of the study is also recommended.

Author Response

Reviewer 2

Comments and Suggestions for Authors

  1. Abstract: Although 4 patients were studied, as indicated in the methods, patient 4 was excluded.  Therefore, only data from 3 patients is reported. Please clarify in the abstract.
  2. Please provide reference for the following statement: “AE attacks are unpredictable and occur episodically upon release of the main mediator of the attack, namely bradykinin, resulting from hyperactivation of the contact system lacking its main control protein .”
  3. For the statement “Indirect costs have been estimated at $16,108, including reduced work- 56 place productivity, reduced income, missed work and travel, and lack of childcare [9],” please indicate if this is yearly costs or per lifetime, etc.
  4. Suggest adding a few sentences or paragraph in the introduction to highlight that bradykinin modulates cardiac vagal tone.A few references of interest:https://www.ncbi.nlm.nih.gov/pmc/articles/PMC5798458/https://pubmed.ncbi.nlm.nih.gov/9277549/
  5. For this paragraph “The aim of this work was to evaluate the cardiac ANS by means of HRV, i.e. the 78 analysis of the spontaneous fluctuations of heart period, usually approximated by the RR 79 interval [15], in C1-INH-HAE patients during an attack and in the hours preceding”, please provide a sentence why that might be important to understand in patients with C1-INH-HAE. 
  6. Results–Figure 1, please indicate/clarify that the data are pooled from the average of multiple measurements from 3 individuals.  In other words, please indicate the n value or provide the Ftest
  7. In Figure 2, it is interesting that the male patient showed a different pattern compared to the females.  Although it is limited in number, it is something the authors may wish to discuss.
  8. Figure 2, please also list the number of measurements that were averaged per patient to generate the graphs.
  9. Discussion—As highlighted above, include a few sentences on potential sex differences that may be present and implications would strengthen the discussion.
  10. Providing a paragraph highlighting the limitations of the study is also recommended.

Response to reviewer 2

We would like to thank the Reviewer 2 for having carefully read our manuscript and for his/her helpful suggestions.

  1. Correct, we changed the number of patients in the abstract from four to three (line 22).
  2. Thank you, the suggested reference was added (ref 2)
  3. Thank you, we specified that the indirect cost is “per year” (line 56)
  4. As suggested, a sentence was added in the “Introduction” (Line 74-75) + ref 13 and ref 23.
  5. We provided the requested sentences in the Introduction (lines from 64 to 67 and from 84 to 86)
  6. We indicated the number of patients analyzed in the text (n=3, line 190) and in Figure 1 title.
  7. Thank you for the helpful observation. We added a comment about possible gender differences in the “Limitations and perspectives” (line 293-295) + ref 28
  8. We changed the legend of figure 2 (line 213). Figure 2 represents the mean over PRE and ONGOING periods of the data shown in Figure 4, i.e. the mean of the four hours of the PRE and four hours of the ONGOING phase of the attack
  9. We added a comment about gender differences in the “Limitations and perspectives” (line 293-295) + ref 28
  10. Absolutely, we added the paragraph (lines 282-295).

Round 2

Reviewer 1 Report

I appreciate the effort from the authors in reply to all my concerns and congratulate their interesting work. I have no more questions.

Reviewer 2 Report

The authors have addressed my previous concerns and the manuscript is now strengthened and improved.